# Clinical Update on Patient-Controlled Analgesia for Acute Postoperative Pain

**DOI:** 10.3390/pharmacy10010022

**Published:** 2022-01-27

**Authors:** Cyrus Motamed

**Affiliations:** Department of anesthesia, Gustave Roussy Cancer Campus, 94080 Villejuif, France; cyrus.motamed@gustaveroussy.fr

**Keywords:** acute pain, postoperative pain, intravenous morphine, patient-controlled analgesia, PCA

## Abstract

Patient-controlled analgesia (PCA) is an effective method for controlling acute pain, including postoperative pain in adults and in children from five years of age, pain resulting from labor, trauma, or other medical situations, or chronic and malignant pain. The treatment consists of a mini-computer-controlled infusion pump permitting the administration of on-demand, continuous, or combined doses of analgesic (mainly opioid) variations in response to therapy, which allows pain to be significantly controlled. Intravenous (IV)-PCA minimizes individual pharmacodynamics and pharmacokinetic differences and is widely accepted as a reference method for mild or severe postoperative pain. IV-PCA is the most studied route of PCA; other delivery methods have been extensively reported in the literature. In addition, IV-PCA usually voids the gap between pain sensation and analgesic administration, permitting better recovery and fewer side effects. The most commonly observed complications are nausea and vomiting, pruritus, respiratory depression, sedation, confusion and urinary retention. However, human factors such as pharmacy preparation and device programming can also be involved in the occurrence of these complications, while device failure is much less of an issue.

## 1. Introduction

Patient-controlled analgesia (PCA) has been used since the early 1970s to relieve multiple categories of pain, including acute, such as postoperative or labor pain, or chronic, such as palliative care or cancer pain [1,2]. The goal of PCA is to efficiently deliver pain relief at a patient’s preferred dose and schedule by allowing them to administer a predetermined bolus dose of medication on-demand at the press of a button [3]. Boluses can be administered alone or coupled with a continuous background infusion of opioids using a dedicated pump.

According to the revised international association for the study of pain definition [4] the new definition of pain is the following: An unpleasant sensory and emotional experience associated with, or resembling that associated with, actual or potential tissue damage. Acute pain is a type of pain that lasts less than 3 months and is generally related to soft tissue injury or damage, such as cuts. It gradually resolves as the injured tissues heal. One of the most common types of acute pain is postoperative pain, which arises in the aftermath of surgery.

Although significant improvement has been made in the anticipation and management of postoperative pain in recent decades, a non-negligible percentage of patients might still have moderate to severe postoperative pain [5].

Patient-controlled analgesia (PCA) preceded by initial intravenous titration is an effective strategy for postoperative analgesia, as it may rapidly provide an adequate analgesic dose upon arrival at the postoperative care unit (PACU).

While intravenous and epidural administration remain the most commonly used modes of PCA, several alternative modes are also available in the clinical setting. These alternative routes of administration include oral, transdermal, inhaled and intranasal, each with its own potential benefits or risks [3].

## 2. Methods

This paper update is a narrative review of only opioid IV-PCA (essentially morphine) for clinicians involved in the management of acute postoperative pain. The search strategy used three electronic databases: PubMed (MEDLINE/Index Medicus), the Cochrane (Controlled Trials Register) and Google Scholar, which were searched for pertinent studies and reviews published between January 2010 and December 2021; animal studies were excluded. Search items included postoperative pain, PCA, patient-controlled analgesia, opioids, parenteral opioids, opioid side effects and morphine. When new updated information was not available older studies were referenced in some areas.

## 3. Results/Discussion

Surgery can result in moderate or severe pain, which can initiate complications in the postoperative period after the procedure. Although anticipation of postoperative pain and its management has been positive for decades, a significant proportion of patients still have high pain scores with non-optimal pain relief [3,4,5].

### 3.1. Predictors of Postoperative Pain

Predictors of postoperative pain and analgesic consumption have been assessed in several studies [6,7,8,9,10,11,12]. The severity of postoperative pain is multifactorial and complex [13]. These factors include preexisting pain, anxiety, age, type of surgery, need for information, genetic factors and history of smoking. Type of surgery, age and psychological distress are the main predictive factors for postoperative analgesic consumption [6,7]. In addition, a practical approach is also reported to be useful by assessing pain score on venous cannulation before surgery [14].

### 3.2. Indications and Benefits of PCA

PCA indications are large, since many surgeries yield high pain scores. Several risk factors for high postoperative pain scores have been established [6,7]. Nearly all types of surgery that yield high postoperative pain can be treated with IV-PCA in the acute phase; examples of such surgeries include spine and other complex orthopedic surgeries, such as knee and hip [15], open abdominal [16] and non-ambulatory laparoscopic surgery [17]; and thoracic [18], major cervicofacial and reconstruction surgeries [19], including breast surgeries [20]. In addition to initial severe postoperative pain, some surgeries can also yield chronic postoperative pain such as cardiac [21], knee [22] shoulder [23] and hip pain [24]. Therefore, special attention should be paid to these surgeries to propose the most efficient technique for patients, which might not be IV-PCA in some circumstances (Table 1).

It is generally recognized that the benefits of PCA are adaptation to individual patients’ needs by giving patients control over their own pain, relatively fast onset of action, predictable drug delivery and side effects, and high scores of satisfaction, despite suboptimal postoperative pain relief in some specific surgeries [18,25].

**Table 1 pharmacy-10-00022-t001:** Options for components of multimodal postoperative pain therapy for commonly performed surgeries.

	PCA IV and/or Other Systemic Therapies *	Side Specific Infiltration or Block with or without Regional Catheters **	Neuraxial Anesthetic Techniques ***
Thoracotomy/thoracoscopy	++	Paravertebral block ++	+++
Laparotomy	++	Infiltration catheters TAP block +	+++
Laparoscopy	+++	Infiltration ++	+
Hip	+++	++	++
Knee	++	+++	++
Shoulder/upper arm	++	+++	
Spinal fusion	+++		++
Cesarean section	++	TAP block ++	+++
Breast surgery	++	Paravertebral block +++	
CABG	+++		
Cervicofacial surgery	+++	+++ When indicated	

* Opioids/non-steroidal anti-inflammatory drugs/Gabapentin or pregabalin/IV ketamine/; ** Block such as femoral, fibular and paravertebral; TAP = transverse abdominal plain block using local anesthetics; *** Epidural with local anesthetics (with or without intrathecal opioid). Most of these surgeries are also reported to yield chronic post-surgical pain [9,10,21,26,27,28,29,30,31,32]. +: moderate indication, ++: acceptable indication,+++: good indication.

### 3.3. Interindividual Variability

Multiple studies involving morphine for postoperative analgesia display a wide range of inter- or intra-individual variability in morphine requirements but also in plasma morphine concentrations [33,34,35,36]. By adapting to patients’ needs, PCA is highly compliant with interindividual needs.

### 3.4. PCA Concept

The effectiveness of PCA is mostly related to the old concept of “minimal effective analgesic concentration” (MEAC), defined as the smallest plasma concentration of morphine at which the pain is relieved [37]. To achieve this concentration, a preliminary titration is necessary, while further adjustments with automatic bolus administered by the patient permit navigation in what is generally described as an analgesic corridor. Trespassing the upper limit of this corridor results in opioid side effects, while exiting the lower limit of this corridor results in the inefficiency of pain relief [38] (Figure 1).

To reach this target, a controlled and progressive increase in the plasma level of opioids is necessary, which is achieved by titration. Titration means that the drug is administered as a bolus of small doses [38]. Titration of morphine in the operating room during wound closure or PACU is generally the first step of IV opioid management. Morphine by titration in PACU requires previous pain assessment by caretakers, generally a numeric verbal scale less than 3–4 on an 11-grade scale (0 no pain, 10 worst ‘maximal’). Titration provides relatively rapid analgesia and the ability to adapt the dose to interindividual variability requirements until the establishment of a clinically acceptable analgesia pattern is obtained; however, on some occasions, another opioid can be used before adequate pain relief is achieved [39]. 

Morphine, a hydrophilic agent and hydromorphone, remains the most commonly used drug for postoperative analgesic titration compared to lipophilic opioids such as fentanyl, sufentanil, alfentanil, or remifentanil [39], as these opioids have a faster onset but also a shorter duration of action.

The maximum concentration after a bolus injection is around six minutes, explaining the delay between further titration injection or the “lock out” period necessary while PCA is used [36,40]. During this lockout time, the PCA pump does not permit further delivery, permitting each bolus to reach the peak effect before the next bolus reducing the risk of overdose.

IV morphine titration allows the dose to be adapted to the patients’ needs and can provide reliable immediate relief of postoperative pain after a wide range of surgical interventions in both young and elderly patients [38]. Titration needs individual adaptation, as the subsequent injection should consider also what has already been administered. Factors affecting early morphine requirements include ethnicity, emergency, major surgery, long-lasting surgery and high pain score upon arrival to PACU [41]. 

It should be emphasized that morphine titration to alleviate acute postoperative pain might not always be effective or possible due to the early appearance of side effects or other complicating factors, such as tachyphylaxis; therefore, it is appropriate to define an alert dose for titration to use alternative methods to alleviate pain [38].

### 3.5. Side Effects of IV-PCA

Most side effects of opioids administered via PCA are related to opioids, such as nausea and vomiting, sedation, apnea, hypoxemia, hypoventilation, pruritus and postoperative delirium (POD).

The ventilatory depressant effects of morphine are the most serious side effects. Although they can be patient-related, they are mostly related to a default in preparation, prescription, or administration, as well as a device failure.

A recent comparison of the side effects of different opioids given in an equianalgesic dose revealed that no significant or clinically relevant difference should be expected in terms of nausea, vomiting and pruritus when comparing different types of opioids [42].

Considering POD, a recent study did not find a difference in its incidence between morphine IV-PCA and fentanyl patient-controlled epidural analgesia (PCEA). After a propensity score matching patient characteristics, it was concluded that POD occurs regardless of the route and dose of opioid administration [43].

Naloxone is reported to decrease the overall incidence of opioid side effects. A pooled analysis study examining IV naloxone (either as a continuous infusion or IV-PCA) revealed a decrease in pruritus and nausea with no increase in pain scores. Overall, the use of IV naloxone is not associated with any significant changes in opioid consumption nor with the risk of sedation or emesis [3,44]. In other more recent study, low and ultra-low doses such as 0.25 µg/kg are also reported to reduce pain intensity due to morphine consumption, pruritus and nausea [45,46]; finally, Pieters [47] found that a high dose naloxone of 0.5 mg/kg/h was not more efficient in reducing opioid side effects, despite partially reversing the analgesic effect of opioid by yielding an increase in demand on postoperative day 2.

While IV-PCA can be used in most types of surgery, it may not provide the best quality of analgesia, especially in dynamic conditions. For example, in open abdominal surgery, epidural analgesia has been considered the gold standard for perioperative analgesia and provides significantly better pain relief both at rest and in dynamic situations [25]. However, after the implementation of enhanced recovery after surgery (ERAS) protocols more than a decade ago and a shift from open to laparoscopic surgery, the advantage of epidural anesthesia has diminished, requiring the use of other opioids and methods, such as intravenous ketamine, peripheral nerve blocks, continuous wound infiltration, intrathecal morphine, intravenous and non-invasive PCA [48].

Although the literature praising PCA-IV for its efficiency in controlling multiple entities of postoperative pain is abundant, according to systematic reviews, the quality of pain relief as determined by assessment of pain intensity scores was only slightly superior to specific non-PCA technique-controlled parenteral opioid (iv, IM, SC) regimens; in two meta-analyses [33,34], PCA is reported to provide superior quality of analgesia with only moderate to low evidence. In addition, despite involving a slightly higher dose of opioid consumption, no increase in opioid-induced side effects is disclosed. Modern surgeries support the use of multimodal regimens in many situations.

As effective and safe alternatives to traditional PCA and with the added benefits of being non-invasive, easy to use and making early patient mobilization possible, newer PCA systems may complement multimodal pain management or replace certain regimens in hospitalized patients with acute postoperative pain [49] (Table 2).

Explaining the PCA pump system to patients is generally performed before surgery at the anesthetic consultation; however, re-instruction after surgery is reported to be effective for optimizing PCA to increase the quality of analgesia. In this way, patients use this technique more efficiently, especially when multiple variables, including dynamic pain, are integrated [50].

In terms of patient’s satisfaction, PCA is again reported to have an edge over non-PCA methods, which does not necessarily mean that the quality of analgesia is superior to the alternative technique [51].

### 3.6. Comparison of Different PCA Medications

Although several opioids have been studied for use in IV-PCA, according to systematic reviews, there is not enough significant clinical evidence to consider the relevant superiority of other opioids on postoperative analgesia compared to morphine IV [33,34] (Table 2 and Table 3).

### 3.7. Adding Ketamine to PCA Morphine

Two recent papers [55,56] investigated the combination of ketamine and opioid IV-PCA, confirming the conclusion of a previous meta-analysis whereby the addition of ketamine to opioid PCA slightly improves postoperative pain intensity and PONV and decreases cumulative morphine consumption. Nevertheless, Matsota et al. found tramadol to be superior compared to ketamine as an adjuvant to morphine IV-PCA [56].

### 3.8. Human-Related Issues and Side Effects

A major human issue associated with adverse effects is non-anticipated/authorized administration of IV bolus doses of an analgesic by either family, friends, or hospital staff, known as PCA by proxy [57]. In a database error analysis, 460 out of 6069 PCA errors resulted in patient harm or death. Twelve out of the 460 errors were attributed to PCA by proxy, with one patient death resulting from nurse-activated PCA [58]. Errors produced by proxy administration programming might also contribute to death from PCA [58,59,60]. In addition, cassette preparation error might yield variation in morphine infusion preparation, leading to inaccuracy of infusion delivery to patients [61].

Multiple and/or successive individual or collective errors can combine to result in a serious adverse event. Both equipment design and human factors play a critical role at multiple levels in generating serious adverse events [62]. On the other hand, PCA can sparsely hide symptoms related to other painful pathologies, such as myocardial infarction, thromboembolism, or acute limb ischemia, which implies vigilance and careful evaluation of opioid consumption in the postoperative period [63].

### 3.9. PCA in Children

Morphine titration is appropriate pain management in pediatric patients because of the limitation in dosage and decreasing the incidence of adverse events. Nevertheless, interindividual variation is also present in this population.

Opioid infusions in children are useful, but adequate dosing and titration require careful supervision. Nevertheless, morphine titration in pediatrics has limitations, as pain scores are not always obtainable in the PACU and the validity of self-reported pain intensity scores may be unreliable in children with developmental and neurological disabilities, thereafter affecting the analgesic protocol and necessitating close monitoring [47,48]. 

Morphine PCA is used for children above 4–6 years of age experiencing painful surgery, such as scoliosis and pectus excavatum repair [64,65]. In addition, for those children unable to handle or understand the concept, PCA by proxy (mainly administered by nurses) remains a safe and efficient method of pain administration for this population, except for children suffering from developmental and neurological disabilities [64,66,67].

### 3.10. PCA for Elderly and Frail Patients

Elderly patients represent a large and rapidly increasing proportion of surgical patients. Management of postoperative pain should focus on minimizing adverse effects.

In normal elderly patients, titration should not be much different; however, for patients older than 90 years and frail patients, a protocol using boluses of 1–2 mg with a greater time interval and a limitation of the total dose is recommended [68].

A retrospective study involving over 10,575 patients comparing older and younger patients using IV fentanyl PCA concluded that, when IV-PCA was used for postoperative pain control, a higher percentage of younger patients required rescue analgesics, while elderly patients required more rescue antiemetics. The addition of ketorolac or ramosetron to the PCA of young and elderly patients can be effective in preventing rescue analgesic or antiemetic use [69]. On the other hand, the cost effectiveness of PCA for elderlies has also been questioned in a small group of orthopedic patients detecting under-use of the device, thus questioning the adequate pre- or postoperative explanations to this category of patients [70]. 

### 3.11. PCA in Obese Patients

Obese patients with obstructive sleep apnea (OSA) are at an increased risk of upper airway obstruction and require close monitoring. Intermittent IV injections of morphine boluses can be used in these patients. Obesity prolongs the elimination half-life of lipophilic drugs such as fentanyl and sufentanil; however, morphine may be used cautiously in patients with sleep apnea syndrome (OSA) without non-invasive positive pressure [18]. Indeed, the lipophilic properties of morphine may result in different tissue concentrations, in addition to a significant decrease in clearance of its glucoronide metabolites [71], in obese patients, placing them at higher risk of respiratory depression. 

PCA, without a background infusion, is a safe and effective method of analgesic delivery. However, if patients are known to have OSA, more intensive monitoring is recommended [72].

### 3.12. PCA in Chronic Pain Patients

Opioid-tolerant patients require additional adjustments when using IV-PCA as the main source of analgesia. Opioid-tolerant patients have significantly higher opioid requirements compared to opioid-naïve patients in the intra- and postoperative period [73]; therefore, adequate postoperative analgesic anticipation is highly recommended [5]. Initial postoperative titration might be difficult using a rapid titration strategy. Using remifentanil in combination with traditional titration effort might help to decrease the time to achieve clinically acceptable levels of pain relief in a better condition [39], while, postoperatively, these patients may often require additional IV continuous infusion doses.

### 3.13. Recent Modalities of PCA Administration 

New modalities of PCA administration are currently under investigation. Lee et al. described a variable feedback infusion rate plus demand in comparison to the constant baseline infusion model plus demand in spinal fusion surgery, whereby the variable model used significantly less analgesic and had significantly less demand than the constant base model; however, there was no difference in side effects [74]. Recently, Jung et al. also described a similar model in laparoscopic cholecystectomy patients whereby the cumulative morphine requirements were significantly less in the optimized background infusion mode, with no difference in the rate of antiemetic use. More studies of different type of surgery are needed before generalizing these modes of administration [75].

### 3.14. Other Factors Influencing Quality of Pain Management

Besides adequate titration, non-pharmacological factors could play a part in optimum pain relief, such as preoperative teaching, adequately addressing patients’ concerns about pain assessment and information about pumps. Adequate knowledge of these factors should help in better clinical care and outcome. 

### 3.15. Limitation of This Study 

Limitations of this study include a bias selection, as the quality of the studies was not classified, and their interpretation, which might have been biased by the own experience of the author.

## 4. Conclusions

Despite the lack of new evidence, IV-PCA administration of opioids, especially morphine, has only a light superiority for quality of analgesia compared to non-PCA opioid-based analgesic regimens. Nevertheless, morphine IV-PCA, as a component of multimodal analgesia for acute postoperative pain, remains a well-accepted/adopted technique by patients and health care providers, resulting in a high level of satisfaction, while the incidence of side effects is equivalent to non-PCA opioid-based techniques. This modality of analgesia will still be valid and useful for the coming years, pending careful selection of patients with appropriate indications.

## Figures and Tables

**Figure 1 pharmacy-10-00022-f001:**
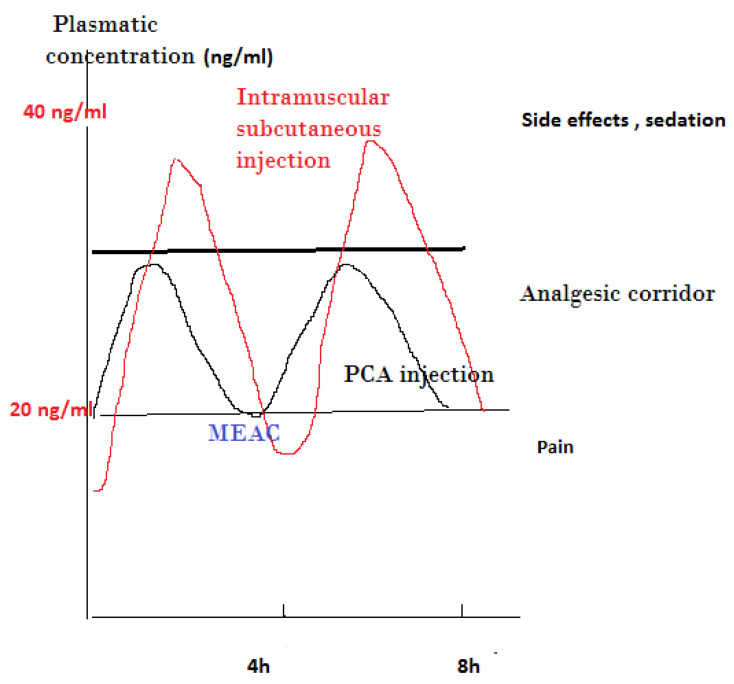
Presentation of analgesic corridor concept: subcutaneous or intramuscular injection vs. PCA IV injection of lower dose of opioid. MEAC, minimum effective analgesic concentration.

**Table 2 pharmacy-10-00022-t002:** Bolus dose and lockout period with different opioids medications.

Analgesic	Bolus Dose	Lockout Period (Minutes)
Morphine	1 mg	5–10
Fentanyl	10 µg	5–10
Hydromorphone	0.25 mg	5–10
Remifentanil	0.5 µg/kg	2
Sufentanil	5 µg	5–10

**Table 3 pharmacy-10-00022-t003:** Comparison of different opioids when used in a PCA mode [52,53,54].

	Efficiency	Side Effects
Oxycodone	As potent as morphine	May have fewer severe side effects
Hydromorphone		Higher incidence of CNS side effects, excitation at higher dose
Fentanyl	High potency +, may require more need for basal infusion rate	Lesser incidence of respiratory depression in comparison to morphine, but more programming errors
Sufentanil	High potency ++, high therapeutic index, more predictable profile, more need for basal infusion	Lower incidence of PONV in comparison to fentanyl
Tramadol	Ten times less potent than morphine	More PONV in some type of surgeries (e.g., spinal fusion)
Remifentanil	Very short duration, studies mainly in labor	Higher respiratory depression, less satisfaction in comparison to epidural analgesia

CNS, central nervous system; PONV, postoperative nausea and vomiting. +: moderate indication, ++: acceptable indication.

## Data Availability

Not applicable.

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
