# Peer review of "Clinical Update on Patient-Controlled Analgesia for Acute Postoperative Pain"

_pharmacy, 2022, doi:10.3390/pharmacy10010022_

Round 1

Reviewer 1 Report

Abstract - 

Introduction - Would recommend adding reference for how acute pain is defined since different readers may not use the same definition.

Methods - Would recommend including search terms used.

Indications and benefits of PCA - greater detail on what type of surgeries are most likely to benefit from post-operative use of PCA and which surgical techniques are more likely to result in chronic pain.

PCA concept - Greater detail on how the titration is needed and that each patient has a minimal effective analgesic concentration that allows for pain relief as long as the serum concentration of the opioid is kept in this range.  Also, hydromorphone is frequently used but not mentioned in this review.

Graph would benefit from concentrations and time scales to further clarify how lock-out period is defined.

Side effects of IV-PCA would be more clear as Side effect of opioids administered via PCA.  Naloxone is discussed briefly as possibly being co-administered with opioid PCA but expansion of this discussion would be beneficial.

A table of which surgeries benefit most from IV-PCA and when other modalities such as epidural or regional block are preferred would help greatly.

Human-related issues and side effects - References 40-46 are older than 2010.  Although this is an important issue, the authors need to clarify that there are no current reports but previously it had been reported 

PCA in children - Would disagree that pain scores are not obtainable in the PACU or may be unreliable.  FLACC score is an accepted and validated methodology in children.

PCA in obese patients - Would add that the lipophilic properties of morphine, etc result in different tissue concentration in obese individuals placing them at higher risk for respiratory depression.

Author Response

Dear  Reviewer 

Thank you for your poitive comments and encouragement; We did our best to comply  with  all of your suggestions 

Introduction - Would recommend adding reference for how acute pain is defined since different readers may not use the same definition.

A reference has been added line 32 and sentence adpated to this definition 

According to the revised international association for the study of pain definition the new definition of pain is: An unpleasant sensory and emotional experience associated with, or resembling that associated with, actual or potential tissue damage

Methods - Would recommend including search terms used.

The search items have been added 

Indications and benefits of PCA - greater detail on what type of surgeries are most likely to benefit from post-operative use of PCA and which surgical techniques are more likely to result in chronic pain.

The different types of surgery has been added (indication of PCA and those yielding chronic pain  in a same table with additional upadted recent references 

PCA concept - Greater detail on how the titration is needed and that each patient has a minimal effective analgesic concentration that allows for pain relief as long as the serum concentration of the opioid is kept in this range.  Also, hydromorphone is frequently used but not mentioned in this review.

Additional phrases explaining titration has now been added 

IV morphine titration allows adaptation of the dose to the patients’ needs and can provide reliable immediate relief of postoperative pain after a wide range of surgical interventions in both young and elderly patients (18). Titration needs individual adaptation as the subsequent injection should consider also what already has been administered. Factors affecting early morphine requirements include ethnicity, emergency, major surgery, long lasting surgery and high pain score upon arrival to PACU(33)

Thank you for your suggestion hydromorphone is now mentionned .

Graph would benefit from concentrations and time scales to further clarify how lock-out period is defined.

Thank you for your suggestion , we inserted  some probablilistic numbers for  time and morphine concentyration to the graph , however this graph is generally  not designed to insist on lock out period   but rather the concept of simplifying  the  understanding of analgesic corridor , in many studies the presentation is without specific concentration or time scales 

for the lock out period we added additional explanation  at the end of the PCA concept page 3

Thank you for your constructive comment.

A table of which surgeries benefit most from IV-PCA and when other modalities such as epidural or regional block are preferred would help greatly.

As suggested a table has now been added partly inspired by the OASP guidelines 

Human-related issues and side effects - References 40-46 are older than 2010.  Although this is an important issue, the authors need to clarify that there are no current reports but previously it had been reported 

Thank you for this suggestion , we added additionnal explanations  in the method section 

. When new updated information was not available older studies were referenced in some areas.

PCA in children - Would disagree that pain scores are not obtainable in the PACU or may be unreliable.  FLACC score is an accepted and validated methodology in children.

. Thank you for your suggestion , we added additional comment to the sentence to refine our phrase 

Nevertheless, morphine titration in pediatrics has, limitations as pain scores are not always obtainable in the PACU and the validity of self-reported pain intensity scores may be unreliable in children with developmental and neurological disabilities, affecting thereafter the analgesic protocol and necessitating close monitoring (47, 48).

PCA in obese patients - Would add that the lipophilic properties of morphine, etc result in different tissue concentration in obese individuals placing them at higher risk for respiratory depression.

Thank you for your suggestion , we inserted the mentionned phrase in addition to a new reference

Indeed lipophilic properties of morphine may result in different tissue concentrations in addition to a significant decrease in clearance of its glucoronide metabolites in obese patients placing them at higher risk of respiratory depression.

Reviewer 2 Report

We need detailed explanations of search strategy, such as what search engine was used and what words were used for literature search, and exclusion and inclusion.   Reference citation number does not match.  Overall, the order and duplication should be revised.

In Result/ Discussion section 3.10 & 3.14, need a more detailed explanation. Please give us a detailed explanation according to the reference.

Author Response

We need detailed explanations of search strategy, such as what search engine was used and what words were used for literature search, and exclusion and inclusion.   Reference citation number does not match.  Overall, the order and duplication should be revised.

Thank you for your suggestion, we rechecked all references and eliminated duplicates , we hope the references are now  adequate 

In Result/ Discussion section 3.10 & 3.14, need a more detailed explanation. Please give us a detailed explanation according to the reference.

Thank you for your suggestion we expanded both of these chapters with new references 

 The 3.10  chapter is now expanded as follows 

Elderly patients represent a large and rapidly increasing proportion of surgical patients. Management of postoperative pain should focus on minimizing adverse effects.

In normal elderly patients, titration should not be much different; however, for patients older than 90 years and frail patients, a protocol using boluses of 1-2 mg with a greater time interval and a limitation of the total dose is recommended (18).

A retrospective study involving over 10575 patients comparing older and younger patients using IV fentanyl PCA concluded that when IV-PCA is used for postoperative pain control, a higher percentage of younger patients may require rescue analgesics, while elderly patients may require more rescue antiemetics. The addition of ketorolac or ramosetron to the PCA of young and elderly patients can be effective in preventing rescue analgesic or antiemetic use (52). On the other hand the cost effectiveness of PCA for elderlies has also been questioned in a small group of orthopedic patients detecting under-use of the device questioning the adequate pre or postoperative explanations to this category of patients(23)

 The 3. 14 is now n°3.13 and has been expanded as it was also requested by reviewer 1 

3.13 :Recent modalities of PCA administration

New modalities of PCA administration are currently under investigation, LEE  et al described a variable feedback infusion rate  plus demand in comparison to  constant baseline infusion model plus demand  in spinal fusion surgery which is  in which the variable model used significantly less analgesic and had significantly less demand  than the constant base model however there was no difference in side effects, recently jung et al also described  similar model in laparoscopic cholecystectomy model in which cumulative morphine requirements was significantly less in the optimized background infusion mode with no difference in the rate of antiemetic use, more studies in different type of surgery are needed before generalizing these modes of administration.

Reviewer 3 Report

Dear Author,

I think the article is written moderately well and can be improved. 

There is certain amount of new information, although the descbed method itself is quite old.

Therefore, athough the author focused on morphine PCA,  first, I would suggest to also review the PCA with other opioids in children and adults. And to compare to morphine PCA, when possible.This would make the article more attractive and useful to the reader. As including only morphine makes the article seem somewhat superficial and "fast-tracked". 

Secondly, I think it is necessary to review novel modalities of opioid PCA administration, e.g. Variable rate PCA infusion. Several articles, concerning variable rate opioid infusion whith PCA are already published. (e.g. A comparison of 2 intravenous patient-controlled

analgesia modes after spinal fusion surgery Constant-rate background infusion versus variable-rate feedback infusion, a randomized controlled trial. 2019, Medicine. 

Third, I would also suggest to avoid generalized statements, when only single articles, describing separate experiments are cited. While, when meta and systemic reviews are cited, this should be clearly stated.

Fourth, I think it is important to include limitations of this narative review before the conclusion. 

Author Response

Therefore, athough the author focused on morphine PCA,  first, I would suggest to also review the PCA with other opioids in children and adults. And to compare to morphine PCA, when possible.This would make the article more attractive and useful to the reader. As including only morphine makes the article seem somewhat superficial and "fast-tracked". 

Thank you for your constructive suggestion , we inserted a new tables with some differences between different medications and added some new references  , but as stated in the first version we believe  there is not enough clinically relevant difference between different drugs to prefer or not prefer a specific drug in comparison to morphine in adults and children for moderate and severe postoperative pain, thank you for your constructive comment

A comparison of 2
intravenous patient-controlled analgesia modes after spinal fusion surgery:

Thank you for your suggestion , we developped an extra paragraph incorporating possible new futures modes of  PCA administration

New modalities of PCA administration are currently under investigation, LEE  et al described a variable feedback infusion rate  plus demand in comparison to  constant baseline infusion model plus demand  in spinal fusion surgery which is  in which the variable model used significantly less analgesic and had significantly less demand  than the constant base model however there was no difference in side effects, recently jung et al also described  similar model in laparoscopic cholecystectomy model in which cumulative morphine requirements was significantly less in the optimized background infusion mode with no difference in the rate of antiemetic use, more studies in different type of surgery are needed before generalizing these modes of administration.

Third, I would also suggest to avoid generalized statements, when only single articles, describing separate experiments are cited. While, when meta and systemic reviews are cited, this should be clearly stated.

Thank you for your advice 

We rechecked our statements and in two occasions we clearly stated according to systematic reviews ..

Although the literature praising PCA-IV for its efficiency in controlling multiple entities of postoperative pain is abundant, according to systematic reviews the quality of pain relief as determined by assessment of pain intensity scores was only slightly superior to specific non-PCA technique controlled parenteral opioid (iv, IM, SC) regimens, with only moderate to low quality evidence that PCA provides superior quality of analgesia

Although several opioids have been studied for use in IV-PCA, according to systematic reviews there is not enough significant clinical evidence to consider the relevant superiority of other opioids on postoperative analgesia compared to morphine IV(33, 34). Table 1 and 2

Fourth, I think it is important to include limitations of this narative review before the conclusion. 

Thank you for your suggestion an additional paragraph has been added incorporating limitation of this narrative review

Round 2

Reviewer 1 Report

Revision has resulted in a much improved manuscript.    Please correct minor English problems to increase readability:  2 examples listed below.

While intravenous and epidural administration remain the most commonly used modes of PCA, several alternative modes have been displayed in the clinical setting.

Also, despite involving a slightly higher dose of opioid consumption, no increase in opioid-induced side effects was noticed in these meta-analyze,

Author Response

Thank you again for your preciuos comment 

Revision has resulted in a much improved manuscript.    Please correct minor English problems to increase readability:  2 examples listed below.

While intravenous and epidural administration remain the most commonly used modes of PCA, several alternative modes have been displayed in the clinical setting.

this sentence has been changed to: While intravenous and epidural administration remain the most commonly used modes of PCA, several alternative modes are also available in  the clinical setting

Also, despite involving a slightly higher dose of opioid consumption, no increase in opioid-induced side effects was noticed in these meta-analyze,

Reviewer 2 Report

This reviewer's comment has been appropriately revised.

Please follow the reference citation format.

[1] 

Lee et al. [74]

Author Response

Thank you again for your constructive comment 

the references have been  apparently reformated by the editor 

Reviewer 3 Report

Dear Author,

Thank you for the revised version of the manuscript - I think it improved significantly.

Here is a couple of minor revisions - corrections which are necessary to make before I can advise it for publication:

  1. Line 208 (track changes vriant) - Tables 1 and 2 or rather 2 and 3, please correct.
  2. Line 316 (Conclusion): I think it is necessary to change the sentence: "Nevertheless, morphine IV-PCA for acute postoperative pain...."  into the sentence: "Nevertheless, morphine IV-PCA as a component of multimodal analgesia for acute postoperative pain..... 

In conclusion, I found the revised manuscript easy, interesting and useful to read. Hope it will be eqully useful for other clinicians as well.

Author Response

Dear reviewever , thank you again for carefully reading  the manuscript and your constructive comment 

all suggested changes were made 

a new read was performed and some sentences were rewrittten 

thank you again for your help